# Pro-Inflammatory and Anti-Inflammatory Interleukins in Periodontitis: Molecular Roles, Immune Crosstalk, and Therapeutic Perspectives

**DOI:** 10.3390/ijms262010094

**Published:** 2025-10-16

**Authors:** Mireya Martínez-García, Enrique Hernández-Lemus

**Affiliations:** 1Departamento de Inmunología, Instituto Nacional de Cardiología Ignacio Chávez, México City 14080, Mexico; mireya.martinez@cardiologia.org.mx; 2División de Genómica Computacional, Instituto Nacional de Medicina Genómica, México City 14610, Mexico

**Keywords:** periodontal disease, inflammation, pro-inflammatory interleukins, anti-inflammatory interleukins, systems immunology

## Abstract

Periodontitis is a chronic, multifactorial inflammatory disease initiated by dysbiotic biofilms and driven by a dysregulated host immune response. Central to its pathogenesis is a complex network of cytokines, particularly interleukins, with both pro-inflammatory and anti-inflammatory roles. This narrative review comprehensively analyzes current knowledge on the molecular biology, cellular sources, immune pathways, and systemic effects of key interleukins in periodontitis. We highlight the dual roles of interleukins such as IL-17 and IL-10, discuss recent advances in understanding their regulatory networks, and explore translational perspectives, including diagnostic biomarkers and cytokine-targeted therapies. Emphasis is placed on dissecting the fine balance between destructive inflammation and protective resolution mechanisms, aiming to inform novel, immune-modulatory treatment strategies for periodontal disease.

## 1. Introduction

Periodontitis is a highly prevalent chronic inflammatory disease that affects the supporting structures of the teeth and remains a major cause of tooth loss in adults worldwide [1]. Although traditionally considered a localized oral condition, increasing evidence indicates that periodontitis is a systemic inflammatory disorder with broad implications for overall health [2,3]. Its pathogenesis is initiated by the accumulation of a polymicrobial biofilm that triggers a local immune response; however, it is the dysregulated host reaction—rather than specific pathogens alone—that drives connective tissue destruction, alveolar bone resorption, and systemic complications [4]. The ulcerated pocket epithelium facilitates the translocation of microorganisms and endotoxins into the bloodstream, leading to transient episodes of bacteremia that may occur several times per day. In addition, locally produced pro-inflammatory mediators such as interleukin (IL)-1 (IL-1), IL-6, tumor necrosis factor (TNF), and prostaglandin E_2_ (PGE_2_) can disseminate into the circulation, amplifying systemic inflammation and potentially affecting distant organs [5].

The microbial component of periodontitis, although necessary for disease initiation, is insufficient to account for disease progression and severity across individuals [6]. Evidence indicates that similar microbial communities can be found in both healthy and diseased periodontal sites [7], underscoring the pivotal role of host factors in shaping disease outcomes [8]. In susceptible individuals, the host–microbe interaction disrupts periodontal tissue homeostasis and activates both innate and adaptive immune pathways, thereby sustaining chronic inflammation [4]. Consequently, the concept of periodontitis as a biofilm-induced, host-mediated disease has gained widespread acceptance and has shifted research efforts toward elucidating the immunological and molecular mechanisms underlying disease pathogenesis [9].

Among the most significant contributors to this dysregulated immune response are cytokines—small proteins that mediate inflammation and coordinate immune cell communication [10,11]. Within this group, interleukins (ILs) play a central role by regulating diverse biological functions, including leukocyte recruitment, activation, differentiation, and tissue remodeling. In periodontitis, ILs orchestrate the transition from a protective immune response aimed at controlling bacterial challenge to a destructive chronic inflammatory state characterized by connective tissue degradation and alveolar bone resorption [12,13,14,15].

Pro-inflammatory interleukins, such as IL-1β, IL-6, IL-17, IL-18, and tumor necrosis factor-alpha (TNF-α), are heavily implicated in the initiation and amplification of periodontal inflammation [16,17,18,19,20,21,22,23,24,25,26,27,28,29,30,31,32,33]. These mediators promote the expression of matrix metalloproteinases (MMPs), osteoclast activation, and the recruitment of neutrophils and macrophages to the periodontal lesion [34]. Among them, IL-17 has emerged as a pivotal cytokine linking adaptive immunity to innate immune activation, thereby creating a feed-forward loop that perpetuates inflammation. In contrast, anti-inflammatory interleukins such as IL-4, IL-10, transforming growth factor-β (TGF-β), and IL-11 suppress excessive immune responses while promoting tissue repair and resolution. The delicate balance between these opposing cytokine forces is essential for maintaining periodontal health, and its disruption underlies disease progression [18,23,35,36,37,38,39,40,41,42,43].

A remarkable feature of interleukins in periodontitis is their pleiotropic and context-dependent behavior. For instance, IL-17 exhibits both protective and destructive roles depending on the timing, intensity, and microenvironment of its expression. Similarly, IL-10 may fail to control inflammation in the presence of persistent microbial dysbiosis or when overwhelmed by pro-inflammatory signals. This complex regulatory network makes interleukins not only central to understanding periodontal pathogenesis but also attractive targets for diagnostic and therapeutic interventions [44,45,46,47].

Beyond their local effects in the periodontium, interleukins can spill into the systemic circulation, contributing to low-grade systemic inflammation (LGI) and increasing the risk for comorbid conditions such as diabetes, cardiovascular disease, rheumatoid arthritis, and adverse pregnancy outcomes [48,49,50,51,52]. Periodontal treatment has been associated with modest improvements in systemic inflammatory markers, supporting the hypothesis that controlling oral inflammation may have broader health benefits. However, further studies are needed to elucidate the causal relationships and mechanistic pathways involved [53,54,55].

Given the centrality of interleukins in both local tissue destruction and systemic immune modulation, a focused review of their roles in periodontitis is timely and warranted. This article aims to synthesize the current understanding of pro- and anti-inflammatory interleukins in periodontal disease, highlighting their cellular sources, signaling pathways, genetic regulation, and clinical relevance. Particular attention was given to the dualistic nature of certain cytokines and their potential as biomarkers and therapeutic targets [56,57,58].

This narrative review explores the immunobiology of pro- and anti-inflammatory interleukins in periodontal inflammation, aiming to bridge the gap between basic molecular research and translational clinical applications. By examining how the cytokine network is regulated—or dysregulated—throughout disease progression, we highlight current evidence and identify promising avenues for precision-medicine approaches in periodontology and related fields.

### Literature Search Strategy

To optimize the thematic coverage of this review and ensure transparency regarding potential sources of bias, a structured search strategy was performed. Specifically, we built systematic queries for the PubMed/MEDLINE, Scopus, and Web of Science databases to identify the relevant literature published between January 1994 and June 2025. Our search terms combined periodontal disease and interleukin-related concepts using Boolean operators—(*periodontitis* OR *periodontal disease*) AND (*interleukin* OR *IL-1* OR *IL-6* OR *IL-10* OR *IL-17* OR *pro-inflammatory cytokines* OR *anti-inflammatory cytokines*)—to maximize sensitivity and specificity.

We included English-language studies conducted in humans or relevant animal models that examined the molecular roles of pro- or anti-inflammatory interleukins in the pathogenesis, immune crosstalk, or therapeutic modulation of periodontitis. Both primary research articles and high-quality narrative or systematic reviews were considered, whereas non-peer-reviewed reports, conference abstracts without full text, and studies unrelated to periodontal inflammation or cytokine biology were excluded. When synthesizing the evidence, we prioritized primary experimental and clinical studies for mechanistic insights and quantitative associations, while review articles were used to contextualize findings and highlight consensus where primary data were limited.

## 2. Overview of Immune Responses in Periodontitis

The immune response in periodontitis is initially mounted as a protective mechanism against microbial invasion, but its chronic and dysregulated nature ultimately drives tissue destruction (see Figure 1A,B). As the lesion progresses, neutrophil- and T helper 1 (Th1)-driven inflammation evolves into established stages marked by epithelial downgrowth, B lymphocyte (B cell) and plasma-cell infiltration, and collagen breakdown (Figure 1C). In advanced disease, failure of resolution pathways sustains high levels of IL-1β, IL-6, IL-17, and TNF-α, which amplify receptor activator of nuclear factor κB ligand (RANKL)-mediated osteoclastogenesis and lead to alveolar bone resorption (Figure 1D) [59,60,61].

This pathological progression reflects the continuous interplay between the oral microbiota and the host immune system at the dentogingival interface, where epithelial cells, resident leukocytes, and microbial products interact. Innate responses initiated by epithelial and stromal cells, neutrophils, dendritic cells (DCs), and macrophages produce key mediators (IL-1β, IL-6, IL-8, TNF-α) that not only drive local inflammation but also shape adaptive immunity by directing T lymphocyte (T cell) differentiation into Th1, T helper 2 (Th2), T helper 17 (Th17), or regulatory T cell (Treg) subsets [62,63,64]. In turn, T cell–derived interleukins regulate macrophage polarization and stromal activity, creating a bidirectional cytokine network that integrates innate and adaptive pathways (see Figure 2) [65,66,67].

The adaptive immune system is characterized by antigen-specific responses mediated by T and B lymphocytes (Figure 2 top right). In periodontitis, several T helper (Th) cell subsets play pivotal roles in modulating disease progression. Th1 cells produce interferon-gamma (IFN-γ) and IL-2, promoting cell-mediated immunity and macrophage activation (Figure 2 bottom). Th2 cells secrete IL-4, IL-5, and IL-13, supporting humoral responses and B cell differentiation, often contributing to antibody production. While both Th1 and Th2 responses have been detected in periodontal tissues, neither alone fully explains disease pathogenesis, suggesting a more complex immunological milieu [68,69].

Th17 cells, a more recently characterized T cell subset, have emerged as central mediators of periodontal inflammation. These cells produce IL-17 and IL-22, which promote neutrophil recruitment and amplify pro-inflammatory cytokine production by epithelial cells and fibroblasts. IL-17 also enhances the RANKL-mediated osteoclast differentiation, linking Th17 cell activation directly to alveolar bone resorption. In experimental models, IL-17-deficient animals exhibit reduced bone loss, underscoring the pathogenic role of this cytokine in periodontitis [70,71].

Treg cells, characterized by the expression of forkhead box P3 (FoxP3) and the secretion of IL-10 and TGF-β, counterbalance the activity of pro-inflammatory T cell subsets. Tregs are essential for maintaining immune tolerance and preventing excessive tissue damage. However, their function may be compromised in chronic periodontitis, allowing pro-inflammatory responses to dominate. A reduced Treg/Th17 ratio in periodontal lesions has been associated with more severe disease, highlighting the importance of regulatory circuits in controlling periodontal inflammation [72,73,74,75].

The immune response in periodontitis is not unidirectional but is governed by complex feedback and crosstalk mechanisms between innate and adaptive pathways. Cytokines produced by innate immune cells can shape T cell differentiation, while T cell-derived cytokines in turn modulate the activity of macrophages, fibroblasts, and osteoclasts. This bidirectional communication results in a self-sustaining inflammatory loop, where cytokine redundancy and synergy contribute to chronicity. For instance, IL-1 and IL-6 produced by macrophages enhance Th17 cells polarization, while IL-17 from Th17 cells reinforces macrophage cytokine production and neutrophil recruitment [3,53,56,65].

The immune landscape of periodontitis is thus defined by a dynamic interplay between innate and adaptive components, where protective responses can become pathogenic when dysregulated. Understanding the precise roles of polymorphonuclear neutrophils (PMNs), macrophages, DCs, and T cell subsets offers critical insights into the molecular pathogenesis of periodontal disease. Moreover, the crosstalk and feedback loops within this immunological network are essential considerations for the development of immune-modulatory therapies aimed at restoring tissue homeostasis.

## 3. Pro-Inflammatory Interleukins in Periodontitis

Pro-inflammatory interleukins are central drivers of the immune-mediated tissue destruction observed in periodontitis (Table 1). They act as molecular messengers that amplify and sustain inflammation, recruit and activate immune cells, and promote bone resorption. Among the most studied of these cytokines are IL-1β [32], IL-6 [17,18], IL-17 [23,34,40], IL-23 [22], and TNF-α [33], with each playing distinct but often overlapping roles in the pathogenic cascade. Their overproduction in the periodontal environment shifts the balance from a protective host response to a destructive chronic inflammatory state.

IL-1β is one of the earliest and most potent pro-inflammatory cytokines involved in periodontal disease. Produced primarily by activated macrophages, DCs, and epithelial cells in response to bacterial products, IL-1β drives the expression of MMPs, prostaglandins, and other mediators that degrade the extracellular matrix. It also promotes osteoclast differentiation through RANKL upregulation, directly linking inflammation to alveolar bone loss. Elevated levels of IL-1β in gingival crevicular fluid are consistently associated with disease severity and progression [32].

IL-6 is another multifunctional cytokine that exerts both local and systemic effects in periodontitis. Locally, IL-6 is produced by macrophages, fibroblasts, and epithelial cells in response to bacterial challenge and other cytokines, such as IL-1 and TNF-α [17]. It promotes Th17 cell differentiation, thereby linking innate immune activation to adaptive immunity. IL-6 also enhances the production and activity of RANKL, both in osteoblasts and in periodontal ligament cells, facilitating osteoclastogenesis and bone resorption. Systemically, elevated IL-6 contributes to low-grade inflammation and is implicated in comorbid conditions such as diabetes and cardiovascular disease [71].

TNF-α is a master regulator of inflammatory responses and is produced by a variety of cells, including macrophages, T cells, and fibroblasts. In periodontal tissues, TNF-α increases vascular permeability, promotes leukocyte recruitment, and induces the production of other pro-inflammatory cytokines and MMPs. Like IL-1β, TNF-α upregulates RANKL, promoting osteoclast formation and bone resorption. Its pleiotropic actions ensure that, once initiated, inflammation becomes self-sustaining and resistant to resolution [29,34,38].

IL-17, predominantly produced by Th17 cells, has emerged as a key cytokine in the pathogenesis of periodontitis. IL-17 stimulates the production of chemokines that recruit neutrophils and enhances the expression of pro-inflammatory mediators in epithelial cells, fibroblasts, and macrophages [22]. It also synergizes with IL-1β and TNF-α to amplify local inflammation. Critically, IL-17 promotes RANKL expression and osteoclast differentiation, providing a direct link to alveolar bone loss [71]. Elevated levels of IL-17 have been detected in periodontal lesions, and experimental models have demonstrated reduced bone loss in IL-17-deficient mice, underscoring its pathogenic role.

IL-23 plays a crucial role in maintaining the Th17 phenotype by supporting the survival and expansion of IL-17-producing cells. Secreted by antigen-presenting cells in response to bacterial products, IL-23 ensures the chronicity of Th17-mediated responses in periodontal tissues [22]. The IL-23/IL-17 axis is therefore central to sustaining the inflammatory milieu that characterizes periodontitis, representing an attractive target for therapeutic intervention.

Other pro-inflammatory interleukins, including IL-18 and IL-12, also contribute to periodontal pathogenesis. IL-18 acts as an IFN-γ–inducing factor, enhancing Th1 responses and promoting the secretion of MMPs, further driving tissue destruction [76]. IL-12, while playing protective roles in pathogen clearance, also amplifies Th1 responses and has been implicated in inflammatory bone loss when dysregulated. These cytokines interact in a complex network with chemokines and other inflammatory mediators to sustain and exacerbate periodontal inflammation [26].

The actions of these pro-inflammatory interleukins are not isolated but highly interconnected. Their signaling pathways exhibit significant redundancy and synergy, creating robust positive feedback loops that perpetuate inflammation. For example, IL-1β and TNF-α upregulate IL-6 and IL-17 production, while IL-17 enhances the expression of IL-1β, TNF-α, and chemokines. This interconnected network ensures that, once established, the inflammatory response is self-reinforcing and resistant to resolution, contributing to the chronicity of periodontitis [26,30].

As we have seen, pro-inflammatory interleukins serve as central orchestrators of periodontal tissue destruction, linking microbial challenge to innate and adaptive immune activation, extracellular matrix degradation, and bone resorption. Understanding their individual and collective roles offers critical insights into the pathogenesis of periodontitis and highlights potential targets for therapeutic interventions aimed at modulating host responses to restore periodontal health.

## 4. Anti-Inflammatory Interleukins and Resolution

While pro-inflammatory interleukins initiate and sustain immune activation in periodontitis, a complementary group of anti-inflammatory interleukins plays a vital role in limiting tissue damage and promoting resolution (Table 2). These cytokines act to suppress excessive immune responses, downregulate pro-inflammatory signaling, and restore tissue homeostasis. In the healthy periodontium, anti-inflammatory interleukins are essential for maintaining immune balance in the face of constant microbial exposure. In periodontitis, however, their regulatory capacity may be overwhelmed or dysregulated, contributing to the persistence of inflammation and tissue destruction [56,77,78].

IL-10 is among the most potent anti-inflammatory cytokines and is produced by Tregs, macrophages, and DCs. IL-10 inhibits the synthesis of pro-inflammatory cytokines such as IL-1β, TNF-α, IL-6, and IL-17, and suppresses antigen presentation and T cell activation. In periodontal tissues, IL-10 expression is associated with reduced inflammatory burden and protection against bone loss. Genetic polymorphisms in the *IL10* gene have been linked to susceptibility to periodontitis, underscoring its central role in modulating disease risk and progression [79,80,81,82].

TGF-β is another key regulator of immune homeostasis and tissue repair. Produced by Tregs, epithelial cells, and fibroblasts, TGF-β inhibits T helper cell activation, suppresses macrophage cytotoxic activity, and promotes the differentiation of Tregs. It also stimulates extracellular matrix production and fibroblast proliferation, contributing to tissue regeneration. In the context of periodontitis, TGF-β may help counteract destructive inflammatory cascades, although its role is complex and context-dependent. Dysregulation of TGF-β signaling has been implicated in impaired resolution and fibrosis in chronic periodontal lesions [74,83,84].

IL-4, secreted primarily by Th2 cells, is involved in the suppression of Th1 and Th17 cells responses [85]. It promotes B cell maturation and antibody production while downregulating the expression of IL-1, IL-6, TNF-α, and RANKL [33,71]. IL-4 also inhibits the activity of MMPs, thereby limiting extracellular matrix degradation. Its immunoregulatory effects help to limit tissue damage and control microbial burden. Experimental models have shown that administration of IL-4 reduces inflammation and bone loss in periodontitis [86,87].

IL-13, closely related to IL-4, shares many of its functions and contributes to the downregulation of pro-inflammatory responses. Both IL-4 and IL-13 signal through a common receptor complex and can inhibit macrophage activation and promote M2 macrophage polarization—associated with tissue repair and anti-inflammatory activity. Although less studied in periodontal disease, IL-13 has been shown to limit inflammation and foster resolution in other mucosal tissues, suggesting a potentially protective role in the periodontium as well [87].

IL-11 and IL-27 are additional anti-inflammatory interleukins with emerging relevance in periodontitis [40,88]. IL-11 modulates hematopoiesis and downregulates pro-inflammatory cytokine expression, while IL-27 enhances IL-10 production and inhibits Th17 cell differentiation. By limiting excessive T cell responses, these cytokines may contribute to the restoration of immune homeostasis and offer potential targets for immune-modulatory therapies. Preclinical studies are beginning to explore their use in inflammatory and autoimmune diseases, including periodontitis [88].

Interestingly, IL-17, commonly classified as a pro-inflammatory cytokine, also exhibits regulatory functions under certain conditions. At low levels, IL-17 supports mucosal barrier integrity and microbial clearance without promoting destructive inflammation. Its dual role underscores the importance of cytokine context, concentration, cellular source, and microenvironmental conditions in determining inflammatory outcomes. A balanced IL-17 response may be essential for controlling infection while avoiding tissue destruction, whereas excessive IL-17 leads to pathogenic consequences [89].

The efficacy of anti-inflammatory interleukins is also shaped by their interaction with pro-inflammatory networks. For example, IL-10 can inhibit IL-17 production, and TGF-β modulates the Th17/Treg balance. However, in chronic periodontitis, persistent microbial stimulation and a pro-inflammatory cytokine milieu can override these regulatory mechanisms. This results in a feed-forward loop where resolution signals are dampened and inflammation persists. Enhancing the function or expression of anti-inflammatory interleukins represents a promising strategy to interrupt this cycle and promote resolution [90,91,92].

Anti-inflammatory interleukins are central to modulating the immune response in periodontitis. Their functions include the inhibition of pro-inflammatory cytokine production, suppression of immune cell activation, and promotion of tissue repair. Dysregulation of these cytokines contributes to disease chronicity and tissue destruction. A deeper understanding of their roles, interactions, and therapeutic modulation may pave the way for targeted interventions aimed at restoring periodontal immune balance and preventing systemic inflammatory consequences [93,94].

## 5. Dual and Context-Dependent Roles

While cytokines are often classified as pro-inflammatory or anti-inflammatory, many interleukins involved in periodontitis exhibit strikingly dual and context-dependent roles. Their effects are not inherently fixed but instead depend on concentration, cellular source, microenvironmental signals, and the phase of the immune response. This complexity underlies much of the difficulty in defining pathogenic versus protective immune responses in periodontal disease, highlighting the need for nuanced understanding of cytokine biology.

IL-17 is perhaps the most illustrative example of this duality. Traditionally viewed as a pro-inflammatory mediator that drives neutrophil recruitment, chemokine production, and osteoclastogenesis via RANKL induction, IL-17 is heavily implicated in the tissue destruction seen in periodontitis. However, IL-17 also plays critical protective roles in maintaining mucosal barrier integrity, promoting antimicrobial peptide production, and supporting host defense against extracellular pathogens. Experimental models have shown that IL-17-deficient animals may be more susceptible to uncontrolled bacterial proliferation and infection, underscoring its essential role in immune defense [95].

The paradox of IL-17 highlights how its pathogenicity depends on the balance of local signals. At low, regulated levels, IL-17 promotes homeostasis by controlling microbial communities and maintaining epithelial barriers. In the setting of persistent microbial dysbiosis and chronic antigenic stimulation, however, IL-17 production becomes dysregulated, resulting in sustained neutrophil recruitment, excessive inflammation, and tissue breakdown. This switch from a protective to destructive function demonstrates how the cytokine milieu and the failure of resolution mechanisms transform a beneficial response into pathology [96,97,98].

Other cytokines also display context-dependent behavior. Transforming growth factor-beta (TGF-β) is broadly anti-inflammatory, suppressing T cell activation and promoting Treg differentiation. However, under certain conditions, TGF-β acts synergistically with IL-6 to drive the differentiation of naïve CD4^+^T cells (CD4^+^ T lymphocytes) into Th17 cells. This functional plasticity means that the same cytokine can promote either resolution or chronic inflammation depending on the inflammatory context, highlighting the importance of the local cytokine environment in shaping immune outcomes [99,100,101].

IL-6 itself is another interleukin with pleiotropic functions. While generally pro-inflammatory and essential for Th17 cells’ differentiation, IL-6 also exerts anti-inflammatory effects by stimulating IL-10 production and promoting acute-phase responses that can help limit infection. Its role in periodontitis is complex, as elevated IL-6 levels have been associated with disease severity and systemic inflammation, yet IL-6 also participates in the initial protective immune responses to bacterial invasion. The net effect of IL-6 signaling thus depends on the timing, duration, and local immune environment [99,100,101,102,103,104].

The balance between Th17 and Treg cells represents a broader example of immune duality in periodontitis [72]. Th17 cells produce IL-17 and IL-22, driving protective antimicrobial responses but also contributing to inflammation and bone loss when unchecked [22,24,105]. Tregs, conversely, suppress excessive inflammation via IL-10 and TGF-β, maintaining tissue homeostasis [72,73,75].

Furthermore, cellular sources and target cell types also influence cytokine effects (See Figure 3). For example, IL-10 produced by Tregs suppresses macrophage and dendritic cell activation, but macrophage-derived IL-10 may have a distinct role in controlling local tissue inflammation [106,107]. Similarly, epithelial cells responding to IL-17 may enhance barrier defenses in health but contribute to destructive chemokine production during disease. This cell-type specificity adds another layer of complexity to cytokine function in the periodontium [22].

Understanding these dual and context-dependent roles is essential for developing effective therapeutic strategies. Interventions that indiscriminately suppress cytokine signaling may risk impairing protective immunity and barrier integrity. Instead, therapies aimed at modulating cytokine balance—enhancing anti-inflammatory pathways while selectively limiting pathogenic signaling—hold greater promise. Such precision approaches require a deep understanding of when, where, and how interleukins exert their effects during periodontal disease progression [11,108,109].

In periodontitis, this interleukin network is characterized by functional plasticity, redundancy, and context-dependence. Far from simple linear pathways, these cytokines act in complex, dynamic networks that determine whether immune responses are protective or destructive. Deciphering these nuances is extremely relevant to gain further knowledge on disease pathogenesis and develop targeted, immune-modulatory therapies that restore periodontal health without compromising essential host defense [110,111,112].

It must be recalled that most human data are cross-sectional and show associations between cytokine levels and disease severity; they do not prove causation. By contrast, gain- and loss-of-function models (e.g., IL-17 knockout mice) provide stronger evidence of causal involvement. Furthermore, confounders such as smoking, diabetes, and genetic background substantially shift the cytokine milieu and may bias associations. Future studies should integrate these variables to clarify causative pathways.

## 6. Genetic and Epigenetic Regulation

The immune response in periodontitis is not only shaped by microbial challenge and local tissue environment but also by the host’s genetic and epigenetic background, which influences cytokine expression and activity. Variations in genes encoding interleukins and their receptors can modulate individual susceptibility to disease, the severity of inflammation, and responsiveness to treatment. Understanding these genetic and epigenetic factors is critical for explaining inter-individual differences in disease presentation and for developing personalized therapeutic strategies [113,114,115,116].

Genetic polymorphisms in interleukin genes have been extensively studied in relation to periodontitis. Polymorphisms in the *IL1* gene cluster, particularly *IL1A* and *IL1B*, have been associated with elevated IL-1β production and increased risk for severe periodontitis in certain populations. Carriers of these polymorphic alleles may exhibit a hyper-inflammatory phenotype, characterized by exaggerated cytokine responses to bacterial challenge [117,118]. Such genetic variants have been proposed as markers for identifying high-risk individuals who might benefit from targeted preventive strategies [36].

Similarly, *IL6* gene polymorphisms can influence systemic and local levels of IL-6, affecting the inflammatory response in the periodontium. Certain *IL6* promoter variants are associated with higher basal and inducible IL-6 expression, which may predispose individuals to chronic inflammation and more severe tissue destruction [13]. Studies have also investigated polymorphisms in the *IL10* gene, which encodes the anti-inflammatory cytokine IL-10. Variants associated with reduced IL-10 production may limit the host’s ability to suppress pro-inflammatory responses, tilting the balance toward unchecked tissue damage.

Beyond single-gene polymorphisms, genome-wide association studies (GWAS) have begun to identify novel loci associated with periodontitis risk. While many GWAS hits implicate immune-related pathways, several point to interleukin signaling networks, highlighting the importance of cytokine regulation in disease susceptibility. However, GWAS findings often show modest effect sizes and significant population heterogeneity, emphasizing the polygenic and multifactorial nature of periodontitis [119,120,121].

Epigenetic regulation adds another layer of complexity to interleukin expression in periodontal disease. DNA methylation, histone modifications, and non-coding RNAs can modulate gene transcription without altering the underlying DNA sequence. For example, the hypermethylation of *IL10* promoter regions may reduce IL-10 expression in periodontal tissues, impairing anti-inflammatory responses. Similarly, the hypomethylation of *IL6* promoters may enhance IL-6 expression, favoring pro-inflammatory signaling [115,122].

Histone modifications also influence the accessibility of interleukin gene loci to transcriptional machinery. The acetylation of histones near *IL1B* or *IL6* promoters can increase gene expression in response to microbial challenge. Conversely, histone deacetylation can repress anti-inflammatory gene expression. These dynamic chromatin changes enable fine-tuned regulation of cytokine responses to environmental stimuli, allowing the immune system to rapidly adapt to microbial invasion but also risking maladaptive chronic inflammation when regulation fails [113,114,123].

Non-coding RNAs, particularly microRNAs (miRNAs), further regulate interleukin expression post-transcriptionally. For example, miR-146a targets signaling molecules in the NF-κB pathway, limiting IL-1β and TNF-α production. The dysregulation of miRNAs in periodontitis has been linked to exaggerated pro-inflammatory cytokine production and impaired resolution mechanisms. Such epigenetic regulators represent potential biomarkers and therapeutic targets for modulating cytokine-driven inflammation in periodontitis [124,125,126,127].

## 7. Systemic Effects and Comorbidities

While periodontitis is fundamentally a local inflammatory disease of the periodontium, its impact extends well beyond the oral cavity. Growing evidence supports a bidirectional relationship between periodontitis and systemic health, mediated in part by the spillover of inflammatory mediators—including interleukins—into the systemic circulation (Figure 4). These cytokines can contribute to LGI, creating a pro-inflammatory state that may exacerbate or predispose one to various chronic diseases [1].

IL-1β, IL-6, TNF-α, and IL-17, key pro-inflammatory interleukins elevated in periodontal tissues, can enter the bloodstream and stimulate systemic inflammatory responses [15]. IL-6, in particular, is a major driver of hepatic acute-phase protein production, including C-reactive protein (CRP), a marker of systemic inflammation and cardiovascular risk. Elevated systemic IL-6 levels have been observed in patients with periodontitis, linking local oral inflammation to broader inflammatory burden in the body [3,128].

One of the most robust associations exists between periodontitis and diabetes mellitus. Chronic periodontal inflammation can impair glycemic control via elevated systemic levels of IL-1β, IL-6, and TNF-α, which interfere with insulin signaling and promote insulin resistance. Conversely, hyperglycemia in diabetic patients enhances pro-inflammatory cytokine production and impairs resolution pathways, creating a vicious cycle. Periodontal treatment has been shown to modestly improve glycemic control, highlighting the clinical relevance of targeting oral inflammation in diabetes management [51,129,130].

Periodontitis has also been linked to cardiovascular diseases (CVD), such as atherosclerosis and coronary artery disease. Pro-inflammatory interleukins originating from periodontal lesions may contribute to endothelial dysfunction, monocyte activation, and plaque development in arterial walls. IL-1β and IL-6 can promote pro-thrombotic states and vascular inflammation, while IL-17 has been implicated in atherogenesis through its effects on endothelial cells and smooth muscle proliferation. This systemic inflammatory burden may partly explain the epidemiological link between poor periodontal health and increased cardiovascular risk [131,132].

Rheumatoid arthritis (RA) represents another condition with strong mechanistic ties to periodontitis. Both diseases are characterized by chronic inflammation, tissue destruction, and similar cytokine profiles—including elevated IL-1β, TNF-α, and IL-17. Periodontal pathogens such as Porphyromonas gingivalis may induce citrullination of host proteins, potentially triggering autoimmune responses in genetically susceptible individuals. Systemic inflammation originating in the periodontium may therefore exacerbate joint inflammation in RA, supporting the concept of a shared inflammatory axis [133,134,135,136,137].

Pregnancy outcomes are also influenced by periodontal inflammation. Elevated systemic levels of IL-1β, IL-6, and TNF-α correlate with adverse pregnancy outcomes, including preterm birth and low birth weight. These cytokines may promote placental inflammation and disrupt normal fetal development. Although interventional studies have produced mixed results, some suggest that periodontal treatment during pregnancy can reduce systemic inflammation and improve birth outcomes, reinforcing the systemic relevance of periodontal health [138,139,140].

The role of anti-inflammatory interleukins, such as IL-10 and TGF-β, is equally important in systemic contexts. The adequate production of these cytokines can mitigate the systemic spread of inflammation from periodontal sites. However, genetic polymorphisms or epigenetic modifications that reduce IL-10 expression may increase susceptibility to both periodontitis and its systemic complications. Restoring the balance between pro- and anti-inflammatory interleukins thus holds promise for improving both oral and overall health [141,142].

## 8. Diagnostic and Therapeutic Implications

The central role of interleukins in the pathogenesis of periodontitis has significant diagnostic and therapeutic implications. As upstream regulators of immune and inflammatory responses, interleukins not only reflect disease activity but also influence its trajectory and response to treatment. Leveraging their diagnostic and prognostic potential, as well as targeting their signaling pathways, offer promising strategies for advancing periodontal care [143,144,145].

Several pro-inflammatory interleukins, notably IL-1β, IL-6, TNF-α, and IL-17, have been proposed as biomarkers for periodontal disease. These cytokines are consistently elevated in the gingival crevicular fluid (GCF), saliva, and serum of individuals with active periodontitis. Their levels often correlate with clinical parameters such as probing depth, attachment loss, and bleeding on probing. Among them, IL-1β and IL-6 have shown particular promise in distinguishing between health, gingivitis, and periodontitis [11,146,147]. The development of chairside assays for these interleukins could improve early detection and enable risk stratification for disease progression.

Anti-inflammatory cytokines, such as IL-10 and TGF-β, may also serve as diagnostic tools, particularly in identifying individuals with an impaired resolution response or regulatory imbalance. The ratio between pro- and anti-inflammatory interleukins (e.g., IL-6/IL-10 or IL-17/TGF-β) could function as a composite biomarker, offering more nuanced insight into the immune profile of the periodontal lesion and systemic inflammatory status [81].

From a therapeutic standpoint, the modulation of interleukin signaling represents a frontier in periodontology (Figure 5). Conventional mechanical debridement and antimicrobial therapy remain the foundation of treatment, but host-modulatory therapies (HMTs) aim to alter the exaggerated inflammatory response that drives disease. Agents targeting interleukins or their signaling pathways—either directly (e.g., monoclonal antibodies) or indirectly (e.g., inhibitors of upstream transcription factors)—are under investigation for use as adjuncts in periodontal therapy [11,108,148].

Monoclonal antibodies against IL-1β (e.g., canakinumab) or TNF-α (e.g., infliximab, etanercept), already in clinical use for rheumatoid arthritis and other inflammatory diseases, hold theoretical potential for controlling periodontal inflammation. Similarly, biologics that inhibit the IL-23/IL-17 axis may suppress Th17-mediated tissue destruction and have been shown to be beneficial in models of periodontal disease. However, systemic cytokine inhibition carries the risk of immunosuppression and opportunistic infections, underscoring the need for targeted delivery systems or localized formulations [149,150,151].

Promoting endogenous resolution pathways represents an alternative therapeutic strategy. For example, boosting IL-10 or TGF-β activity through gene therapy, cytokine delivery, or small-molecule agonists may enhance immune regulation and tissue-healing. Additionally, lifestyle modifications (e.g., smoking cessation, dietary interventions, improved glycemic control) have been shown to modulate cytokine profiles and may enhance the efficacy of periodontal therapy through indirect immunomodulation [152,153,154].

Recent advances in precision periodontology emphasize the integration of cytokine-based diagnostics and therapeutics with genetic and epigenetic profiling. Personalized treatment regimens based on interleukin expression patterns, host genotypes, and microbiome composition could improve outcomes and reduce overtreatment. As technologies such as point-of-care biosensors, multiplex cytokine assays, and RNA-based diagnostics become more accessible, they are likely to play a growing role in individualized periodontal care [155,156].

Recent evidence from inflammatory arthritis underscores how cytokine modulation affects not only inflammatory cascades but also the fundamental survival programs of immune cells. Vomero and collaborators [157] demonstrated that anti-TNF therapy in RA patients reduces autophagy and enhances apoptosis in lymphocytes, changes that correlated with improved clinical outcomes. These findings reveal that TNF blockade reshapes immune cell homeostasis beyond the simple suppression of pro-inflammatory cytokines, actively dismantling the persistence mechanisms that sustain chronic inflammation. This concept is directly relevant to periodontitis, a disease characterized by prolonged immune activation and defective resolution. Similarly to RA, the excessive TNF signaling and heightened survival of pathogenic lymphocyte subsets in periodontal tissues may contribute to the chronicity and tissue destruction typical of advanced disease stages.

Complementary insights emerge from the work of Riitano and coworkers [158], who reported dysregulation of the Wnt signaling pathway—marked by elevated DKK1, Wnt5a, and β-catenin—in RA and psoriatic arthritis, correlating with disease activity and bone erosion. The positive association between β-catenin and IL-6 highlights the tight coupling between inflammatory cytokine networks and bone metabolism. This mechanism resonates with periodontitis, where IL-6 and other pro-inflammatory interleukins promote osteoclastogenesis and alveolar bone resorption. Taken together, these studies reinforce the notion that inflammatory arthritis and periodontitis share convergent pathogenic pathways involving TNF signaling, autophagy dysregulation, and Wnt-mediated bone remodeling. Targeting these shared axes—whether through the modulation of autophagy, inhibition of TNF or IL-6 signaling, or fine-tuning of Wnt pathways—represents a promising translational strategy for controlling chronic inflammation and preventing bone loss in periodontal disease.

In the present context, interleukins offer a valuable window into the immune dysregulation underlying periodontitis and represent both diagnostic biomarkers and therapeutic targets. Incorporating cytokine profiling into clinical practice has the potential to transform the diagnosis, risk assessment, and management of periodontitis, moving the field toward a more personalized and immune-modulatory model of care.

It must be stressed, however, that although biologics targeting IL-1β, TNF-α, and the IL-23/IL-17 axis have shown efficacy in systemic inflammatory diseases and in experimental periodontitis models (Table 3), randomized controlled trials in periodontal patients are still scarce (for a list of currently registered clinical trials see Appendix A).

Most evidence to date derives from animal studies or extrapolation from RA or psoriasis cohorts. Systemic administration carries risks of immunosuppression and opportunistic infections, and optimal dosing for periodontal indications has not been established. Targeted local delivery systems (e.g., gels, microspheres, or gene therapy vectors) are under investigation to minimize systemic exposure while achieving high gingival concentrations. Until the results of well-designed clinical trials are available, these agents should be considered experimental adjuncts with a cautious risk–benefit profile. Future research should focus on validating cytokine-based tools in diverse populations and developing safe, effective interleukin-targeted interventions [159,160,161].

## 9. Scope and Limitations of This Review

As a narrative synthesis, our review is limited by potential publication bias and heterogeneity in the study designs, sampling matrices, and assay methods. Many cited studies are cross-sectional, with small sample sizes and limited adjustment for confounders such as smoking or metabolic comorbidities. Moreover, evidence for interleukin-targeted therapies in periodontitis remains largely preclinical. These limitations should be considered when interpreting the translational implications.

## 10. Conclusions and Future Directions

Periodontitis is increasingly recognized as a complex immunoinflammatory disease where the balance between pro-inflammatory and anti-inflammatory interleukins critically shapes disease initiation, progression, and resolution. Far beyond simple host responses to microbial challenge, the cytokine networks active in periodontal tissues orchestrate the delicate interplay between protection and destruction. Understanding the molecular and cellular roles of interleukins is essential for appreciating the heterogeneity of disease expression and the links between oral and systemic health [162,163].

Pro-inflammatory interleukins—including IL-1β, IL-6, TNF-α, and IL-17—have emerged as central mediators of tissue breakdown, bone resorption, and chronic inflammation. Their redundancy and synergy create robust positive feedback loops that sustain disease. Conversely, anti-inflammatory interleukins such as IL-10, TGF-β, IL-4, and IL-13 function as critical regulators that limit excessive immune responses and promote healing. The disruption of this dynamic equilibrium is a defining feature of periodontitis pathology [56,93,95].

One of the most compelling insights from recent research is the context-dependent duality of several interleukins. For example, IL-17 plays indispensable protective roles in mucosal defense but becomes pathogenic when chronically elevated. Similarly, TGF-β can suppress inflammation or drive Th17 differentiation depending on the surrounding cytokine milieu. These pleiotropic functions underscore the importance of nuanced, phase-specific approaches to cytokine modulation rather than indiscriminate suppression [24,72,85].

Genetic and epigenetic factors add additional layers of complexity, predisposing some individuals to hyper-inflammatory or hypo-regulatory cytokine profiles. As genome-wide association studies and epigenomic analyses continue to evolve, they offer the promise of personalized risk assessment and precision interventions tailored to each patient’s immunogenetic landscape. Incorporating these insights into routine clinical practice remains a significant but exciting challenge [113,114,116,120].

The systemic implications of periodontal cytokine dysregulation are increasingly clear. Elevated circulating interleukins may lead to low-grade systemic inflammation and are mechanistically linked to comorbid conditions such as diabetes, cardiovascular disease, rheumatoid arthritis, and adverse pregnancy outcomes. This recognition reinforces the view that managing periodontal inflammation has health benefits extending far beyond the oral cavity.

Looking forward, several areas warrant focused research. These include the development of biomarkers capable of accurately reflecting disease activity and predicting progression; the refinement of host-modulatory therapies that selectively restore immune balance without compromising protective defenses; and the creation of delivery systems for local, controlled cytokine inhibition or enhancement. Additionally, studies exploring how lifestyle factors, nutrition, and microbiome modulation intersect with interleukin networks provide important insights into integrative treatment strategies.

From a translational perspective, integrating cytokine-based diagnostics and therapeutics into clinical workflows will require robust validation studies, cost-effectiveness analyses, and clinician education. Innovations such as point-of-care assays, salivary biosensors, and digital health tools will likely play key roles in operationalizing these advances. Multidisciplinary collaboration among periodontists, immunologists, geneticists, and primary care providers will be essential to realize the full potential of cytokine-focused interventions.

In conclusion, interleukins occupy a central role at the intersection of microbial challenge, host response, and systemic health in periodontitis. Deciphering their functions and regulatory networks not only advances our understanding of periodontal pathogenesis but also opens new frontiers for diagnosis, prevention, and treatment. Future efforts to harness this knowledge hold the promise of more precise, effective, and patient-centered care in periodontology and beyond.

## Figures and Tables

**Figure 1 ijms-26-10094-f001:**
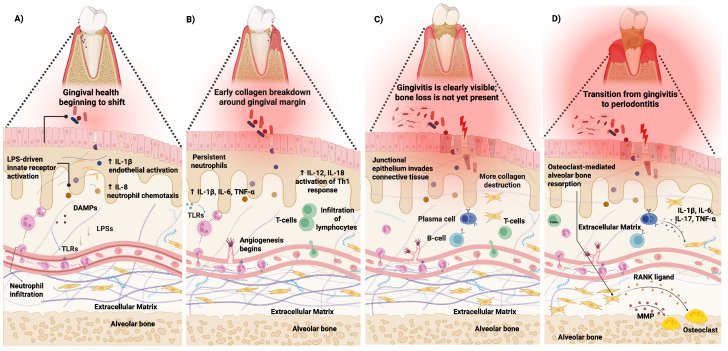
Stages of periodontal lesion development. (**A**) Initial lesion: Within 2–4 days of biofilm accumulation, lipopolysaccharides (LPS) and damage-associated molecular patterns (DAMPs) activate Toll-like receptors (TLRs), triggering IL-1β and IL-8 release, endothelial activation, and neutrophil recruitment into the gingival sulcus. (**B**) Early lesion: By 4–7 days, persistent neutrophil infiltration is accompanied by increased IL-1β, IL-6, and tumor necrosis factor-alpha (TNF-α). Angiogenesis begins, and lymphocytes infiltrate the connective tissue. IL-12 and IL-18 drive T helper 1 (Th1) polarization. (**C**) Established lesion: With sustained inflammation, the junctional epithelium proliferates into connective tissue. B and plasma cells accumulate, T-cell activity persists, and collagen destruction accelerates, though bone resorption is not yet apparent. (**D**) Advanced lesion: Transition to periodontitis is marked by receptor activator of nuclear factor κB ligand (RANKL)-mediated osteoclast activation and alveolar bone resorption. Pro-inflammatory cytokines (IL-1β, IL-6, IL-17, TNF-α) and matrix metalloproteinases (MMPs) amplify tissue breakdown, establishing a chronic destructive inflammatory loop. Arrows connect specific cells and interactions (Figure created with Biorender.com).

**Figure 2 ijms-26-10094-f002:**
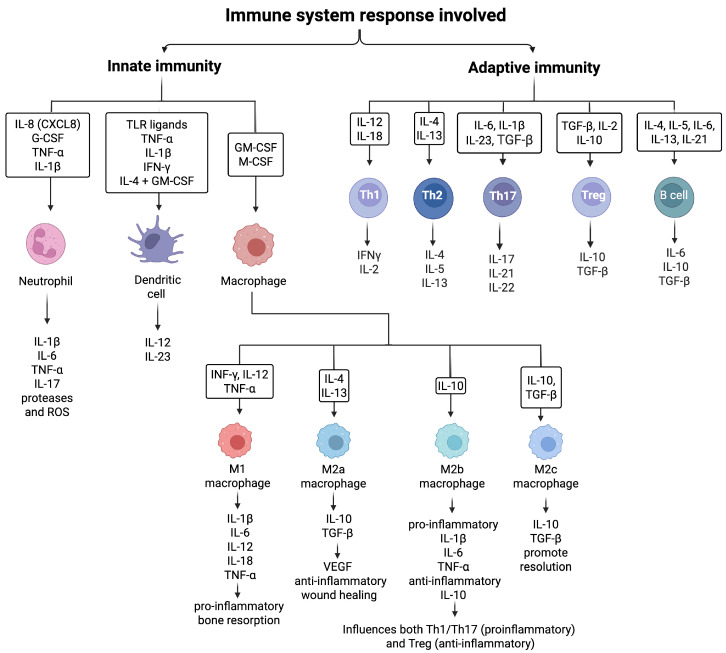
Overview of immune responses and interleukin cross-talk in periodontitis. Schematic representation of the main cellular sources, interleukins, and downstream effects coordinating innate and adaptive immunity in the periodontal environment. Innate signals (IL-1β, IL-6, IL-8, TNF-α, GM-CSF) recruit neutrophils and activate dendritic cells and macrophages, which in turn guide T-cell differentiation. Adaptive subsets (Th1, Th2, Th17, Treg, B cells) secrete distinct interleukins that regulate inflammation, macrophage polarization, and tissue responses. Macrophage plasticity (M1 vs. M2) and the balance between pro-inflammatory cytokines (e.g., IL-17, TNF-α) and anti-inflammatory mediators (IL-10, TGF-β) determine outcomes ranging from receptor activator of nuclear factor κB ligand (RANKL)-driven bone loss to resolution and repair. CXCL8: C-X-C Motif Chemokine Ligand 8; G-CSF: Granulocyte Colony-Stimulating Factor; TNF-α: Tumor Necrosis Factor-alpha; IFN-γ: Interferon-gamma; TLR: Toll-Like Receptor; M-CSF: Macrophage Colony-Stimulating Factor; GM-CSF: Granulocyte–Macrophage Colony-Stimulating Factor; VEGF: Vascular Endothelial Growth Factor; M1: Classically activated macrophage phenotype; M2a: Alternatively activated macrophage subtype; M2b: Regulatory macrophage subtype; M2c: Deactivated macrophage subtype. Arrows connect specific cells and interactions (Figure created with Biorender.com).

**Figure 3 ijms-26-10094-f003:**
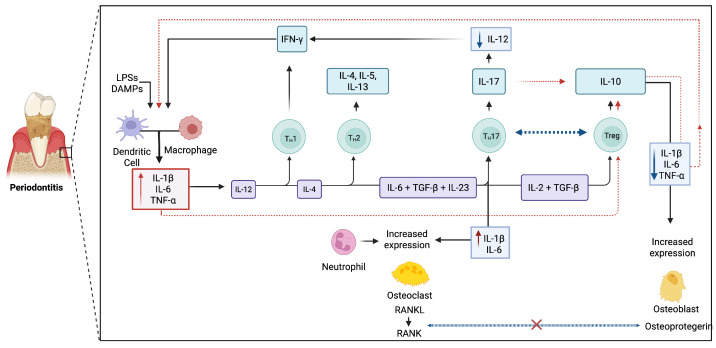
Cross-talk cytokine pathway in periodontitis. A bidirectional interleukin network regulates inflammation and bone remodeling in periodontal disease. Pro-inflammatory mediators (IL-1β, IL-6, IL-17, TNF-α) upregulate RANKL expression, activating the RANK–RANKL pathway and driving osteoclastogenesis and alveolar bone resorption. In contrast, anti-inflammatory cytokines (IL-10, TGF-β, IL-4) limit excessive inflammation and promote tissue repair, with osteoprotegerin (OPG, secreted by osteoblasts) acting as a decoy receptor for RANKL to preserve bone. The Th17/Treg axis is a pivotal regulatory node: Th17-derived IL-17 amplifies destructive inflammation, whereas Treg-associated IL-10 and TGF-β counterbalance tissue damage and restore homeostasis. Several interleukins, including IL-6, TGF-β, and IL-17, exert dual and context-dependent roles, underscoring the complexity of cytokine cross-talk in determining periodontal outcomes. Arrows indicate signaling processes (Figure created with Biorender.com).

**Figure 4 ijms-26-10094-f004:**
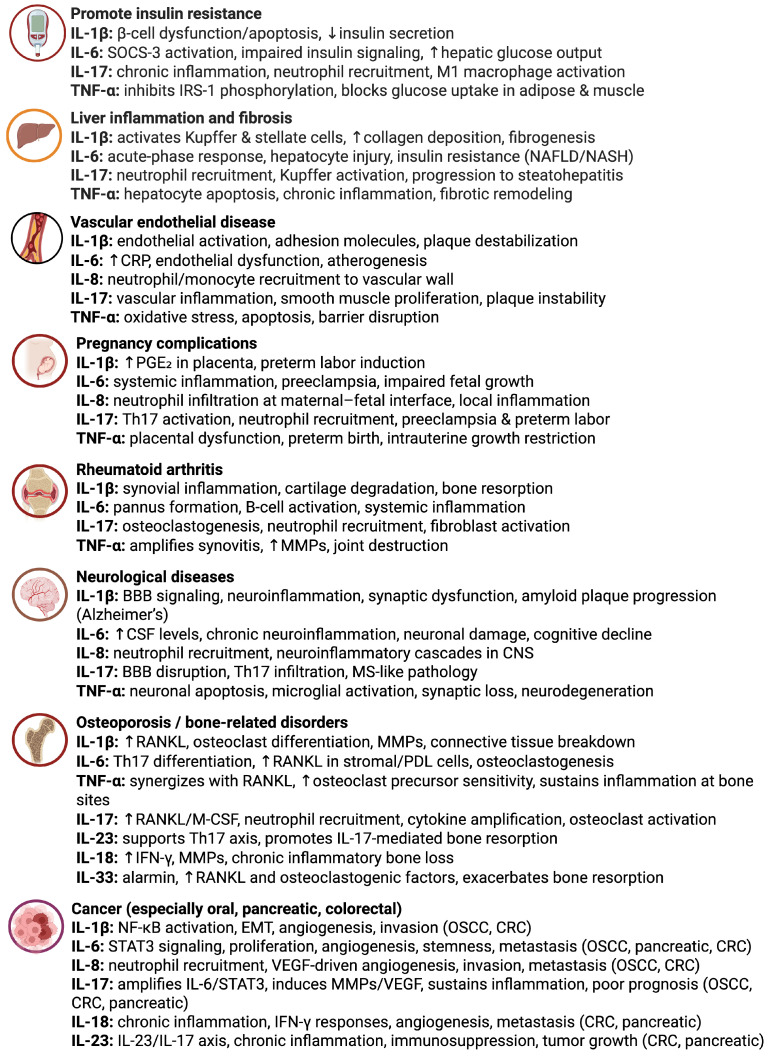
Systemic implications of pro-inflammatory interleukins derived from periodontitis. Chronic periodontitis sustains elevated production of IL-1β, IL-6, IL-8, TNF-α, IL-17, IL-18 and IL-23 creating a persistent inflammatory microenvironment. These interleukins spill into systemic circulation, raise acute-phase proteins such as CRP, and act as common inflammatory mediators linking periodontal inflammation with systemic diseases. ↓: decrease; ↑: increase; IL: interleukin; TNF: tumor necrosis factor; SOCS-3: suppressor of cytokine signaling 3; IRS-1: insulin receptor substrate-1; NAFLD: non-alcoholic fatty liver disease; NASH: non-alcoholic steatohepatitis; CRP: C-reactive protein; PGE_2_: prostaglandin E_2_; MMPs: Matrix Metalloproteinases; BBB: blood–brain barrier; CNS: central nervous system; MS: multiple sclerosis; PDL: periodontal ligament; M-CSF: macrophage colony-stimulating factor; OSCC: oral squamous cell carcinoma; CRC: colorectal cancer; VEGF: vascular endothelial growth factor; IFN: interferon (Figure created with Biorender.com).

**Figure 5 ijms-26-10094-f005:**
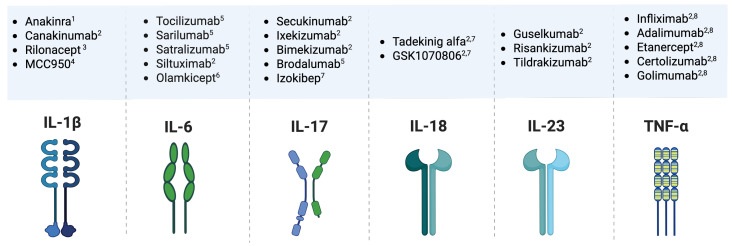
Representative pathway inhibitors targeting key pro-inflammatory interleukins and tumor necrosis factor-alpha (TNF-α) involved in periodontitis and related systemic diseases. **Upper panels** list representative agents classified by mechanism of action, while **lower panels** depict the corresponding targets. These biologic and small-molecule inhibitors act as: ^1^ receptor antagonists; ^2^ neutralizing antibodies; ^3^ trap or decoy receptors; ^4^ inflammasome inhibitors; ^5^ receptor blockers; ^6^ selective trans-signalling inhibitors; ^7^ engineered cytokine binders; and ^8^ receptor decoys for TNF-α. Collectively, these agents modulate the inflammatory axis by neutralizing cytokine–receptor interactions or downstream signalling, holding therapeutic potential to attenuate systemic inflammation secondary to chronic periodontitis (Figure created with Biorender.com).

**Table 1 ijms-26-10094-t001:** Pro-inflammatory interleukins in periodontal disease.

IL	Source	Target	Effect	Relationship with Other Interleukins
IL-1β	Fb, GECs, DCs and Mφ	Endo cells and Osteoclast precursors	Amplifies inflammation, induces MMPs and RANKL; enhances osteoclastogenesis and tissue destruction	Synergizes with TNF-α and induces IL-8
IL-6	Fb, Endo cells, Epi cells and Mφ	Hepatocytes, B cells, T cells and Osteoclasts	Stimulates acute-phase response and promotes osteoclast differentiation	Induced by IL-1β and TNF-α; promotes IL-17 production
IL-8	Fb, Epi cells and Mφ	Neutrophils	Neutrophil recruitment and activation; sustains inflammation and collateral tissue damage	Induced by IL-1β and TNF-α; promotes further IL-8 via neutrophil activation
IL-12	DCs and Mφ	Naïve T cells and NK cells	Promotes Th1 differentiation and IFN-γ production	Works with IL-18 to enhance IFN-γ; antagonizes IL-4 effects
IL-17	Th17 cells	Fb, Epi cells, Osteoblasts and Mφ	Stimulates neutrophil recruitment and induces pro-inflammatory cytokines and MMPs	Induced by IL-6 and IL-23; promotes IL-8 and TNF-α
IL-18	DCs and Mφ	T cells and NK cells	Enhances IFN-γ production, promotes Th1 responses and contributes to tissue damage	Works with IL-12 to promote Th1; synergizes with IL-1β and counter-regulated by IL-10
IL-23	DCs and Mφ	Th17 cells	Stabilizes and expands Th17 responses that drive bone resorption	Upstream of IL-17A/F; induced by TLR/IL-1β signals
IL-33	Fb, Epi cells and Endo cells	Th2 cells and Mast cells	Acts as alarmin, triggers type-2 cytokine production and contributes to tissue inflammation and repair	Induces IL-4, IL-5 and IL-13; interacts with Treg cells

IL:interleukin; IL-17A/F (A/F): IL-17A and IL-17F isoforms; TNF: tumor necrosis factor; IFN: interferon; RANKL: receptor activator of nuclear factor-κB ligand; MMPs: matrix metalloproteinases; Fb: fibroblasts; GECs: gingival epithelial cells; Epi cells: epithelial cells; Endo cells: endothelial cells; DCs: dendritic cells; Mφ: macrophages; NK: natural killer cells; Treg: regulatory T cells; Th1: T helper 1 lymphocytes; Th2: T helper 2 lymphocytes; Th17: T helper 17 lymphocytes.

**Table 2 ijms-26-10094-t002:** Anti-inflammatory interleukins in periodontal disease.

IL	Source	Target	Effect	Relationship with *Other Interleukins*
IL-4	Th2 cells and Mast cells	Naïve T cells and Mφ	Promotes humoral/Th2 responses and suppresses excessive macrophage activation and MMP production	Induced by IL-33; antagonizes IL-12 and IFN-γ
IL-10	Treg cells and Mφ	Mφ, DCs and T cells	Suppresses pro-inflammatory cytokine production and antigen presentation, limiting tissue damage	Inhibits IL-1β, IL-6, and TNF-α
IL-11	Fb and Epi cells	Mφ	Downregulates TNF-α and MMPs, protects connective tissue, and modulates mucosal immunity	Antagonizes IL-1β and TNF-α; acts synergistically with IL-10
IL-13	Th2 cells and Mast cells	Mφ	Induces M2-like macrophage polarization and may limit excessive tissue destruction	Induced by IL-4 and IL-33; antagonizes IL-12
IL-27	DCs and Mφ	Naïve T cells	Suppresses Th17 differentiation, induces IL-10, and promotes a balance between inflammation and tissue healing	Antagonizes IL-6 and IL-17 pathways

IL: interleukin; IFN: interferon; TNF: tumor necrosis factor; DCs: dendritic cells; Mφ: macrophages; Th17: T helper 17 lymphocytes; Th2: T helper 2 lymphocytes; Treg: regulatory T cells; Fb: fibroblasts; Epi cells: epithelial cells; MMPs: matrix metalloproteinases.

**Table 3 ijms-26-10094-t003:** Summary of diagnostic performance of IL-1β, IL-6, and IL-17 in gingival crevicular fluid (GCF) and saliva from periodontitis studies.

Cytokine	Matrix (Periodontitis vs. Control)	Assay/Platform	Effect Size (Fold-Change)	Diagnostic Metrics (AUC, Cut-Off When Available)	Pre-Analytical Notes
IL-1β	GCF (site-specific) [32]; Saliva [155]	ELISA, Helsinki, Finland, multiplex bead immunoassay	∼3–5× higher in active sites vs. healthy [32]; decreases after therapy [19]	Tomás et al., 2017 nomogram including IL-1β yielded AUC ≈ 0.82 for chronic periodontitis [159]	Standardize collection (isolated vs. pooled GCF), avoid saliva stimulation; store at −80 °C
IL-6	GCF, saliva, serum [17,35]	ELISA or Luminex^®^, Austin, TX, USA multiplex	∼2–4× higher in periodontitis vs. controls [17]; reduction parallels clinical improvement [19]	Combined with IL-1β improved discrimination (AUC ≈ 0.80–0.83) [159]	Circadian variation; influenced by smoking/diabetes; matrix choice critical
IL-17	GCF [42]; Saliva (smokers vs. nonsmokers) [94]	ELISA, high-sensitivity multiplex	Detected in 60–80% of diseased vs. <20% of healthy sites [32]; salivary IL-17 rises with disease stage and smoking [92]	Mohammed et al., 2024: IL-17 + IL-10 panel AUC ≈ 0.78 for stage III–IV vs. controls [92]	Freeze–thaw sensitive; single vs. pooled GCF alters values; adjust for smoking status

## Data Availability

No new data were created or analyzed in this study. Data sharing is not applicable to this article.

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
