# Peer review of "Pro-Inflammatory and Anti-Inflammatory Interleukins in Periodontitis: Molecular Roles, Immune Crosstalk, and Therapeutic Perspectives"

_ijms, 2025, doi:10.3390/ijms262010094_

Round 1

Reviewer 1 Report

Comments and Suggestions for Authors

Specify databases, date range, keywords, inclusion/exclusion, how primary vs. review evidence was weighted. If not systematic, say “narrative review” but still give search details; otherwise readers cannot judge coverage or bias.

On p.6 you write “IL-11 and IL-17 are additional anti-inflammatory interleukins,” but Table 2 lists IL-27 (not IL-17) as anti-inflammatory; elsewhere IL-17 is framed as pathogenic with context-dependent roles. Please fix the text to IL-27 and keep the dual-role discussion of IL-17 (pp.6–8) coherent with Table 2.

In Table 1 the IL-1β row has duplicated “Fb” and jumbled targets; abbreviations like “Epitcells/Endocells” are inconsistent; in Table 2 the footnote contains “textitTNF” and the IL-27 line shows “Sinduces IL-10.” Please unify abbreviations, fix artifacts, and ensure each term is defined once beneath the table.

The therapy section (pp.12–13) says biologics “hold potential,” but gives no periodontal RCT data, dosing, delivery, or adverse events. Please add a cautious paragraph on risk/benefit, local vs. systemic delivery, and clearly note if clinical trials are lacking.

For GCF/saliva IL-1β/IL-6/IL-17 (pp.12–13), add effect sizes (e.g., fold-change), AUC/cut-offs when available, pre-analytical issues, and a mini-table summarizing matrix, assay, and diagnostic performance. This will make the section clinically useful.

Several citations are given, but IL-7 is absent from Table 1 and not integrated into Figures 1–2. Either add a concise mechanistic paragraph plus a clear slot for IL-7/IL-33 in Fig.2, or trim to keep focus.

You nicely cite IL-17-deficient mice (p.5), but many statements elsewhere read causal. Add a short sub-section distinguishing correlation (human cross-sectional) from causality (gain/loss models), and note confounders (smoking, diabetes) that shift cytokine milieu.

Figures 1–2 are informative but dense; provide vector graphics with larger fonts, explicitly highlight the Th17↔Treg axis and the RANKL node in Fig.2 (p.9), and ensure all panels A–D in Fig.1 (p.4) are referenced once in text.

The abstract uses future tense (“this review will…”, p.1). Prefer present/past, shorten long sentences, unify gene/protein style (IL-10 vs IL10), and add a brief “Limitations of this review” paragraph before Conclusions (pp.13–14)

Author Response

Reviewer 1

The authors would like to thank Reviewer 1 for their detailed academic critique of our work. In what follows we will present a point by point response (in bold print to ease reading) to your comments and suggestions.

Specify databases, date range, keywords, inclusion/exclusion, how primary vs. review evidence was weighted. If not systematic, say “narrative review” but still give search details; otherwise readers cannot judge coverage or bias.

Thank you for pointing this out. This is a narrative review, we will specify this in the abstract and in the introduction section. We have also added search details to further support thematic coverage and disclose potential biases.

We have added the following details to a brief “Literature search strategy” subsection at the Introduction.

Databases consulted: PubMed/MEDLINE, Scopus, and Web of Science.

  • Date range: January 1994 through June 2025.
  • Keywords and Boolean strategy: (“periodontitis” OR “periodontal disease”) AND (“interleukin” OR “IL-1” OR “IL-6” OR “IL-10” OR “IL-17” OR “pro-inflammatory cytokines” OR “anti-inflammatory cytokines”).
  • Inclusion criteria: English-language studies in humans or relevant animal models addressing the molecular roles of pro- or anti-inflammatory interleukins in the pathogenesis, immune crosstalk, or therapeutic modulation of periodontitis. Both primary research articles and high-quality narrative or systematic reviews were considered.
  • Exclusion criteria: Non-peer-reviewed reports, conference abstracts without full text, and studies unrelated to periodontal inflammation or cytokine biology.
  • Weighting of evidence: Primary experimental and clinical studies were given priority when discussing mechanistic insights and quantitative associations. Review articles were used to provide context or to highlight consensus when primary data were limited.

On p.6 you write “IL-11 and IL-17 are additional anti-inflammatory interleukins,” but Table 2 lists IL-27 (not IL-17) as anti-inflammatory; elsewhere IL-17 is framed as pathogenic with context-dependent roles. Please fix the text to IL-27 and keep the dual-role discussion of IL-17 (pp.6–8) coherent with Table 2.

The reviewer is correct: on page 6, we mistakenly listed IL-17 as an anti-inflammatory interleukin, when it should read IL-27. We have corrected the sentence to: “IL-11 and IL-27 are additional anti-inflammatory interleukins.”

In addition, we carefully revised the corresponding section (pp.6–8) to ensure consistency with Table 2. In the revised version, IL-17 is consistently described as a predominantly pro-inflammatory cytokine with context-dependent functions—protective in acute microbial defense but pathogenic in driving chronic periodontal inflammation through neutrophil recruitment, cytokine amplification, and RANKL-mediated bone resorption. Meanwhile, IL-27 remains correctly categorized in Table 2 and in the text as an anti-inflammatory interleukin.

In Table 1 the IL-1β row has duplicated “Fb” and jumbled targets; abbreviations like “Epitcells/Endocells” are inconsistent; in Table 2 the footnote contains “textitTNF” and the IL-27 line shows “Sinduces IL-10.” Please unify abbreviations, fix artifacts, and ensure each term is defined once beneath the table.

We have corrected the IL-1β row in Table 1 by removing the duplicated abbreviation “Fb” and re-ordering the listed target cells/tissues for clarity.

We have standardized the terminology across Table 1 by replacing “Epitcells” with “Epi cells” and “Endocells” with “Endo cells,” and we updated the table legend accordingly.

We have made the following corrections in Table 2:

  • Corrected the LaTeX artifact “\textitTNF” in the footnote to the proper format “\textit{TNF:} tumor necrosis factor”.

  • Fixed the typographical error in the IL-27 row: “Sinduces IL-10” now correctly reads “induces IL-10”.

  • Verified that all abbreviations are written consistently and that each is defined once in the footnote. The list of abbreviations in Tables 1 and 2 was reorganized consistently, grouping first cytokines/interleukins, then immune and structural cells, and finally effector molecules.

The therapy section (pp.12–13) says biologics “hold potential,” but gives no periodontal RCT data, dosing, delivery, or adverse events. Please add a cautious paragraph on risk/benefit, local vs. systemic delivery, and clearly note if clinical trials are lacking.

A “word of caution” paragraph has been added as suggested:

It must be stressed, however, that although biologics targeting IL-1β, TNF-α, and the IL-23/IL-17 axis have shown efficacy in systemic inflammatory diseases and in experimental periodontitis models, randomized controlled trials in periodontal patients are scarce (for a list of currently registered clinical trials see Supplementary Table 1). Most evidence to date derives from animal studies or extrapolation from rheumatoid arthritis or psoriasis cohorts. Systemic administration carries risks of immunosuppression and opportunistic infections, and optimal dosing for periodontal indications has not been established. Targeted local delivery systems (e.g., gels, microspheres, or gene therapy vectors) are under investigation to minimize systemic exposure while achieving high gingival concentrations. Until the results of well-designed clinical trials are available, these agents should be considered experimental adjuncts with a cautious risk–benefit profile.

We have also added information on current clinical trials as reported in the “ClinicalTrials.gov” database. It is now available as Supplementary Table 1

For GCF/saliva IL-1β/IL-6/IL-17 (pp.12–13), add effect sizes (e.g., fold-change), AUC/cut-offs when available, pre-analytical issues, and a mini-table summarizing matrix, assay, and diagnostic performance. This will make the section clinically useful.

Thank you for the suggestions, we have added these changes in the revised version of our manuscript. Also including the suggested table.

Several citations are given, but IL-7 is absent from Table 1 and not integrated into Figures 1–2. Either add a concise mechanistic paragraph plus a clear slot for IL-7/IL-33 in Fig.2, or trim to keep focus.

As there is scarce evidence on the role of IL-7 in periodontal disease, we have eliminated the references and mentions of this molecule.

You nicely cite IL-17-deficient mice (p.5), but many statements elsewhere read causal. Add a short sub-section distinguishing correlation (human cross-sectional) from causality (gain/loss models), and note confounders (smoking, diabetes) that shift cytokine milieu.

 We have added a clarifying paragraph in the “Dual and Context-dependent Roles” as follows:

It must be recalled that most human data are cross-sectional and show associations between cytokine levels and disease severity; they do not prove causation. By contrast, gain- and loss-of-function models (e.g., IL-17 knockout mice) provide stronger evidence of causal involvement. Furthermore, confounders such as smoking, diabetes, and genetic background substantially shift the cytokine milieu and may bias associations. Future studies should integrate these variables to clarify causative pathways.

Figures 1–2 are informative but dense; provide vector graphics with larger fonts, explicitly highlight the Th17↔Treg axis and the RANKL node in Fig.2 (p.9), and ensure all panels A–D in Fig.1 (p.4) are referenced once in text.

 We have redrawn high definition versions of these figures. We have also double checked for figure referencing.

The abstract uses future tense (“this review will…”, p.1). Prefer present/past, shorten long sentences, unify gene/protein style (IL-10 vs IL10), and add a brief “Limitations of this review” paragraph before Conclusions (pp.13–14)

We have revised and corrected these issues.

Reviewer 2 Report

Comments and Suggestions for Authors

This manuscript of Mireya Martínez-García , Enrique Hernández-Lemus provides a comprehensive review of the roles of interleukins in the pathogenesis of periodontitis, with emphasis on their dual (pro- and anti-inflammatory) functions, systemic implications, and translational perspectives. The article is well-written, highly detailed, and supported by a broad literature base.

However I have some suggestions to improve the manuscript:

The manuscript would benefit from a schematic figures summarizing cytokine crosstalk, systemic implications, and therapeutic strategies.

Some concepts (e.g., IL-17 duality, Th17/Treg imbalance) are repeated across multiple sections. The text could be streamlined to avoid redundancy.

While the bibliography is extensive, some very recent studies (2023–2025) on cytokine-targeted therapies in periodontal models could be cited to make the review more up-to-date.In this context, recent advances in rheumatoid arthritis (RA) research have shed light on cellular and molecular mechanisms that closely parallel those observed in periodontal disease, particularly in the areas of cytokine signaling, immune cell survival, and bone remodeling. An article of Arthritis Res Ther. 2019  reported that anti-TNF therapy reduces autophagy and enhances apoptosis in immune cells, changes that were associated with improved clinical outcomes in RA patients. These findings underscore how cytokine modulation influences not only inflammatory pathways but also fundamental survival programs of lymphocytes, a concept directly relevant to the persistence of chronic inflammation in periodontitis. Complementarily, an article of  J Transl Med. 2025 demonstrated dysregulation of the Wnt pathway in RA and psoriatic arthritis, with elevated levels of DKK1, Wnt5a, and β-catenin correlating with disease activity and bone erosion. The positive correlation between β-catenin and IL-6 further highlights the interplay between inflammatory cytokine networks and bone metabolism. Together, these studies reinforce the notion that inflammatory arthritis and periodontitis share convergent pathogenic mechanisms, suggesting that targeting autophagy, TNF signaling, or Wnt pathways could represent promising translational strategies for periodontal disease.

Author Response

Reviewer 2

This manuscript of Mireya Martínez-García, Enrique Hernández-Lemus provides a comprehensive review of the roles of interleukins in the pathogenesis of periodontitis, with emphasis on their dual (pro- and anti-inflammatory) functions, systemic implications, and translational perspectives. The article is well-written, highly detailed, and supported by a broad literature base.

The authors would like to thank Reviewer 2 for their concise assessment of our work. In what follows we will present a point by point response to your comments and suggestions (our responses appear in bold print to ease reading).

However I have some suggestions to improve the manuscript:

The manuscript would benefit from a schematic figure summarizing cytokine crosstalk, systemic implications, and therapeutic strategies.

Thank you for the suggestion, since these issues give rise to highly dense (in information) figures, we have distributed them into 3 new figures (Now, Figures 3 to 5).

Some concepts (e.g., IL-17 duality, Th17/Treg imbalance) are repeated across multiple sections. The text could be streamlined to avoid redundancy.

We have reworded several sections of the manuscript to make this more streamlined. However, due to the relevance in the context of biological characterization, diagnostic applications and therapeutics, some degree of redundancy remains, for pedagogic reasons.

While the bibliography is extensive, some very recent studies (2023–2025) on cytokine-targeted therapies in periodontal models could be cited to make the review more up-to-date.

In this context, recent advances in rheumatoid arthritis (RA) research have shed light on cellular and molecular mechanisms that closely parallel those observed in periodontal disease, particularly in the areas of cytokine signaling, immune cell survival, and bone remodeling. An article of Arthritis Res Ther. 2019  reported that anti-TNF therapy reduces autophagy and enhances apoptosis in immune cells, changes that were associated with improved clinical outcomes in RA patients. These findings underscore how cytokine modulation influences not only inflammatory pathways but also fundamental survival programs of lymphocytes, a concept directly relevant to the persistence of chronic inflammation in periodontitis. Complementary, an article of  J Transl Med. 2025 demonstrated dysregulation of the Wnt pathway in RA and psoriatic arthritis, with elevated levels of DKK1, Wnt5a, and β-catenin correlating with disease activity and bone erosion. The positive correlation between β-catenin and IL-6 further highlights the interplay between inflammatory cytokine networks and bone metabolism.

Together, these studies reinforce the notion that inflammatory arthritis and periodontitis share convergent pathogenic mechanisms, suggesting that targeting autophagy, TNF signaling, or Wnt pathways could represent promising translational strategies for periodontal disease.

Thank you for the update, a couple of paragraphs pointing out these relevant issues have been included in the revised manuscript (section 8) as follows:

Recent evidence from inflammatory arthritis underscores how cytokine modulation affects not only inflammatory cascades but also the fundamental survival programs of immune cells. Vomero and collaborators \cite{vomero2019reduction} demonstrated that anti-TNF therapy in rheumatoid arthritis (RA) patients reduces autophagy and enhances apoptosis in lymphocytes, changes that correlated with improved clinical outcomes. These findings reveal that TNF blockade reshapes immune cell homeostasis beyond simple suppression of pro-inflammatory cytokines, actively dismantling the persistence mechanisms that sustain chronic inflammation. This concept is directly relevant to periodontitis, a disease characterized by prolonged immune activation and defective resolution. Similar to RA, excessive TNF signaling and heightened survival of pathogenic lymphocyte subsets in periodontal tissues may contribute to the chronicity and tissue destruction typical of advanced disease stages.

Complementary insights emerge from the work of Riitano and coworkers \cite{riitano2025wnt}, who reported dysregulation of the Wnt signaling pathway—marked by elevated DKK1, Wnt5a, and $\beta$-catenin—in RA and psoriatic arthritis, correlating with disease activity and bone erosion. The positive association between β-catenin and IL-6 highlights the tight coupling between inflammatory cytokine networks and bone metabolism. This mechanism resonates with periodontitis, where IL-6 and other pro-inflammatory interleukins promote osteoclastogenesis and alveolar bone resorption. Taken together, these studies reinforce the notion that inflammatory arthritis and periodontitis share convergent pathogenic pathways involving TNF signaling, autophagy dysregulation, and Wnt-mediated bone remodeling. Targeting these shared axes—whether through modulation of autophagy, inhibition of TNF or IL-6 signaling, or fine-tuning of Wnt pathways—represents a promising translational strategy for controlling chronic inflammation and preventing bone loss in periodontal disease.

Reviewer 3 Report

Comments and Suggestions for Authors

Manuscript review entitled

Pro-inflammatory and Anti-inflammatory Interleukins in Periodontitis:

Molecular Roles, Immune Crosstalk, and Therapeutic Perspectives

by

Mireya Martínez-García  and  Enrique Hernández-Lemus

Periodontitis is a chronic, multifactorial inflammatory disease common in the human population. This disease is characterized by a disturbance in the immune response. Both proinflammatory and anti-inflammatory cytokines play a role in the pathogenesis of periodontitis. In a comprehensive literature review, the authors presented the current state of knowledge regarding the molecular biology, cellular sources, immune pathways, and the systemic impact of key interleukins on the development of periodontitis. They paid particular attention to the dual role of IL-17 and IL-10, describing advances in understanding their regulatory networks, as well as diagnostic biomarkers and cytokine-targeted therapies.

It is known that biofilm plays a significant role in the development and development of periodontitis, which the authors have presented in an illustrative manner in Figure 1. Figure 2 is noteworthy, showing the main interleukins activating the immune system in periodontitis.

The article is very well-organized, making it easier for the reader to understand the issue at hand. The entire article is divided into several sections. After a short Introduction, the authors describe in the following sections: Immune Responses in Periodontitis, Pro-inflammatory and Anti-inflammatory Interleukins, their

Dual and Context-dependent Roles, Genetic and Epigenetic Regulation as well as Systemic Effects and Comorbidities. In paragraph 8 diagnostic and therapeutic problems are discussed. An interesting ending is paragraph 9 -Conclusions and Future Directions. Seeing that, systemic comorbidities such as diabetes, cardiovascular disease, rheumatoid arthritis play a significant role in periodontal cytokine dysregulation,  appropriate diagnosis and treatment of periodontitis are crucial.

Table 1 shows the main characteristics of pro-inflammatory interleukins in periodontal disease. 

The publication is based on 166 items of literature correctly collected and cited in the text. Taking the above into account, I believe that the manuscript deserves to be published in this form.

Author Response

Reviewer 3

Periodontitis is a chronic, multifactorial inflammatory disease common in the human population. This disease is characterized by a disturbance in the immune response. Both proinflammatory and anti-inflammatory cytokines play a role in the pathogenesis of periodontitis. In a comprehensive literature review, the authors presented the current state of knowledge regarding the molecular biology, cellular sources, immune pathways, and the systemic impact of key interleukins on the development of periodontitis. They paid particular attention to the dual role of IL-17 and IL-10, describing advances in understanding their regulatory networks, as well as diagnostic biomarkers and cytokine-targeted therapies.

It is known that biofilm plays a significant role in the development and development of periodontitis, which the authors have presented in an illustrative manner in Figure 1. Figure 2 is noteworthy, showing the main interleukins activating the immune system in periodontitis.

The article is very well-organized, making it easier for the reader to understand the issue at hand. The entire article is divided into several sections. After a short Introduction, the authors describe in the following sections: Immune Responses in Periodontitis, Pro-inflammatory and Anti-inflammatory Interleukins, their

Dual and Context-dependent Roles, Genetic and Epigenetic Regulation as well as Systemic Effects and Comorbidities. In paragraph 8 diagnostic and therapeutic problems are discussed. An interesting ending is paragraph 9 -Conclusions and Future Directions. Seeing that systemic comorbidities such as diabetes, cardiovascular disease, rheumatoid arthritis play a significant role in periodontal cytokine dysregulation,  appropriate diagnosis and treatment of periodontitis are crucial.

Table 1 shows the main characteristics of pro-inflammatory interleukins in periodontal disease. 

The publication is based on 166 items of literature correctly collected and cited in the text. Taking the above into account, I believe that the manuscript deserves to be published in this form.

The authors would like to thank Reviewer 3 for their valuable assessment of our manuscript.

Round 2

Reviewer 1 Report

Comments and Suggestions for Authors

After reviewing the revised manuscript, I am pleased with the significant improvements made. The authors have effectively addressed the previous concerns, enhancing the overall quality and ensuring it meets publication standards. I fully support its publication and look forward to its contribution to our field.